# *Epichloë gansuensis* Increases the Tolerance of *Achnatherum inebrians* to Low-P Stress by Modulating Amino Acids Metabolism and Phosphorus Utilization Efficiency

**DOI:** 10.3390/jof7050390

**Published:** 2021-05-17

**Authors:** Yinglong Liu, Wenpeng Hou, Jie Jin, Michael J. Christensen, Lijun Gu, Chen Cheng, Jianfeng Wang

**Affiliations:** 1State Key Laboratory of Grassland Agro-Ecosystems, Center for Grassland Microbiome, Key Laboratory of Grassland Livestock Industry Innovation, Ministry of Agriculture and Rural Affairs, Engineering Research Center of Grassland Industry, Ministry of Education, College of Pastoral Agriculture Science and Technology, Lanzhou University, Lanzhou 730000, China; liuyl2020@lzu.edu.cn (Y.L.); houwp19@lzu.edu.cn (W.H.); gulj@lzu.edu.cn (L.G.); chengch20@lzu.edu.cn (C.C.); 2Key Laboratory of Cell Activities and Stress Adaptations, Ministry of Education, School of Life Sciences, Lanzhou University, Lanzhou 730000, China; jinj2015@lzu.edu.cn; 3Retired Scientist of AgResearch, Grasslands Research Centre, Private Bag 11-008, Palmerston North 4442, New Zealand; mchristensenpn4410@gmail.com

**Keywords:** *Epichloë gansuensis*, low P stress, metabolomics, grass

## Abstract

In the long-term evolutionary process, *Achnatherum inebrians* and seed-borne endophytic fungi, *Epichloë gansuensis*, formed a mutually beneficial symbiosis relationship, and *Epichloë gansuensis* has an important biological role in improving the tolerance of host grasses to abiotic stress. In this work, we first assessed the effects of *Epichloë gansuensis* on dry weight, the content of C, N, P and metal ions, and metabolic pathway of amino acids, and phosphorus utilization efficiency (PUE) of *Achnatherum inebrians* at low P stress. Our results showed that the dry weights, the content of alanine, arginine, aspartic acid, glycine, glutamine, glutamic acid, L-asparagine, lysine, phenylalanine, proline, serine, threonine, and tryptophan were higher in leaves of *Epichloë gansuensis*-infected (E+) *Achnatherum inebrians* than *Epichloë gansuensis*-uninfected (E−) *Achnatherum inebrians* at low P stress. Further, *Epichloë gansuensis* increased C content of roots compared to the root of E− plant at 0.01 mM P and 0.5 mM P; *Epichloë*
*gansuensis* increased K content of leaves compared to the leaf of E− plant at 0.01 mM P and 0.5 mM P. *Epichloë*
*gansuensis* reduced Ca content of roots compared to the root of E− plant at 0.01 mM P and 0.5 mM P; *Epichloë gansuensis* reduced the content of Mg and Fe in leaves compared to the leaf of E− plant at 0.01 mM P and 0.5 mM P. In addition, at low P stress, *Epichloë gansuensis* most probably influenced aspartate and glutamate metabolism; valine, leucine, and isoleucine biosynthesis in leaves; and arginine and proline metabolism; alanine, aspartate, and glutamate metabolism in roots. *Epichloë gansuensis* also affected the content of organic acid and stress-related metabolites at low P stress. In conclusion, *Epichloë gansuensis* improves *Achnatherum inebrians* growth at low P stress by regulating the metabolic pathway of amino acids, amino acids content, organic acid content, and increasing PUE.

## 1. Introduction

Phosphorus (P) is an important macronutrient for plant life cycle, including growth and development, which is involved in a variety of physiology processes. It is significant for the synthesis of some important compounds and regulation of metabolic pathways, such as membrane lipids, nucleic acids, ATP, and phosphorylation intermediates [1]. According to statistics, 30% to 40% of crop yield is limited by phosphorus availability [2]. With the continuous growth of population and the increasing global demand for food and biofuels, the global food supply has gradually become a high-yield agriculture relying on the use of artificial phosphate fertilizer. In modern agriculture, people apply a lot of phosphate fertilizer to enhance the content of soil phosphorus and crop yield [3]. Presently, phosphate rock resources are being consumed at a high speed [4]. Meanwhile, unfortunately, intensive application of mineral phosphate fertilizer causes negative effects on the environment and it has adverse impacts on soil health and the structure, and composition of soil microbial flora [5]. As a result, soil fertility and crop yield decreased gradually [6]. In order to maintain the sustainable development of agriculture and food security under the global climate change and improve the phosphorus absorption and utilization efficiency of plant, breeding grasses with high P efficiency is an efficient strategy to deal with low P availability of soil.

*Achnatherum inebrians* is mainly distributed in the natural grassland of North and Northwest China, including Gansu, Xinjiang, Qinghai, Tibet, and Inner Mongolia. *Achnatherum inebrians* is the host of seed-borne *Epichloë* endophytes, and the infection rate could nearly reach 100% [7]. A large number of studies have confirmed that *Epichloë* endophyte can improve the resistance of host grasses (such as *Achnatherum inebrians, Festuca arundinacea, Festuca sinensis, Hordeum brevisubulatum, Lolium perenne*) to various stresses, including salt stress, drought stress, waterlogging stress, heavy metal stress, and low nitrogen stress [8,9,10]. Moreover, *Epichloë* endophytes can affect plant community, soil nutrients, and soil microbial diversity [11,12,13]. Importantly, *Achnatherum inebrians-Epichloë* symbiosis in China, *Festuca arundinacea- Epichloë*
*coenophia**la* symbiosis in America, and *Lolium perenne- Epichloë festucae var. lolii* symbiosis in New Zealand have become important research fields of endophytic fungi of grasses in the world.

Soil nutrient availability is one of the important factors influencing the effect of endophytes on host grasses. One study have shown that endophyte infection affects the distribution of photosynthetic products in roots and improves the ability of nutrient removal, therefore, increasing the resistance of tall fescue to P deficiency [14]. Besides, recent study confirm that *Epichloë* endophyte plays an important role in promoting *Lolium perenne* growth in low fertility soil by increasing root growth, metabolic activity, biomass, and changing nutrients [15].

In recent years, metabolomics has been widely used to study and solve complex plant nutrition problem. Metabolomics apparently improves understanding of adaptive mechanisms and responses to a variety of alterations to environmental conditions by analyzing a complete set of metabolites [16]. Meanwhile, metabolomics is rapidly emerging as an effective tool to research the tolerance of plant to various stresses, including temperature stress, salt stress, and drought stress [17]. However, although there has been some research on *Epichloë* endophyte improving plant resistance to low P stress, little is known about mechanisms involved in this tolerance to low P stress. Therefore, metabolomics approach is an effective and necessary way to understand *Epichloë gansuensis* increasing *Achnatherum inebrians* tolerance to low P stress. The objectives of our work are to (1) investigate the role of *Epichloë gansuensis* on the dry weight, metal ions content, and metabolites changes of different tissues of *Achnatherum inebrians* at low P stress, (2) demonstrate the important metabolic pathways of leaves and roots in response to *Epichloë gansuensis* for increasing the host tolerance to low P stress, (3) determine the effect of *Epichloë gansuensis* on the content of amino acid in leaves and roots, and phosphorus utilization efficiency at low P stress. Finally, we found specific key metabolic pathways were influenced by *Epichloë gansuensis*, which were beneficial for reprograming the physiology of host grasses to improve the growth of *Achnatherum inebrians* at low P stress.

## 2. Materials and Methods

### 2.1. Achnatherum Inebrians Seedlings Growth and the Different P Concentrations Treatment

The seeds preparation of *Epichloë gansuensis*-infected (E+) and *Epichloë gansuensis*-uninfected (E−) *Achnatherum inebrians* was similar to our previous research with some modifications having been made [11]. Briefly, 12 pots for E+ seedlings and 12 pots for E− seedlings were used in the present experiments, and each pot was filled with vermiculite that was sterilized at 150 °C for 3.5 h in an oven. The size of the pots was as follows, lower diameter: 10 cm; height: 19.5 cm; upper diameter: 18.5 cm. After well-watering the sterile vermiculite, six healthy E+ or E− seeds were sown in each of the E+ and E− pots, respectively. Six E+ or 6 E− pots were assigned to one tray, respectively. The trays were put into a greenhouse, with moisture: 42 ± 2%, light: 16 h/8 h (light/dark), and temperature: 26 ± 2 °C. Seven days after the germination of E+ and E− seeds, three seedlings with similar growth were retained in corresponding E+ and E− pots. E+ and E− seedlings were treated with modified 1/2 Hoagland with two different P concentrations (0.01 mM and 0.5 mM) for 18 weeks. All E+ and E− seedlings were treated with two P concentrations, and each P concentrations had six independent biological replicates. After 18 weeks of the treatment with different P concentrations, the dry weight, the content of metal ions and amino acids, and phosphorus utilization efficiency were determined and the metabolomics experiment was performed by GS-MS.

### 2.2. The Dry Weight of Leaves and Roots in E+ and E− Achnatherum inebrians and Preparation of GS-MS

After 18 weeks of the treatment with the different P concentrations, the roots and leaves of all E+ and E− plants were removed from the vermiculite, then washed with distilled water and collected. Six E+ plants and six E− plants of the different P treatments were dried at 80 °C for the determination of dry weight of leaves and roots. Roots and leaves of other six E+ *Achnatherum inebrians* and six E− *Achnatherum inebrians* were collected at 0.01 and 0.5 mM P, and frozen in liquid nitrogen, immediately stored at −80 °C, which were used for metabolomics assay.

### 2.3. The Determination of C, N, P and Metal ion Contents of E+ and E− Achnatherum inebrians Plants

Dried leaves and roots were crushed in a ball grinder respectively. A part of the samples was used to determine the total organic carbon (TOC) of leaves and roots by K_2_Cr_2_O_7_-H_2_SO_4_ oxidation method [18]. The other part of the samples was used to determine nitrogen (N), P, potassium (K), calcium (Ca), sodium (Na), magnesium (Mg), and iron (Fe) contents of leaves and roots by wet digestion with H_2_O_2_-H_2_SO_4_, at 370 °C for 1.5 h, during which H_2_O_2_ was added every 7 min until the sample in a digestion tube is transparent, then increased the volume to 50 mL, finally the samples were filtered awaiting determination [18]. The N and P contents of leaves and roots were analyzed by automatic batch chemical analyzer (SmartChem 140, France) and the measurement of K, Ca, Na, Mg, and Fe contents was done by an atomic absorption spectrophotometer (Aurora, AI-1200, Canada) according to the description of Han et al. [19].

### 2.4. GC-MS Analysis

In order to perform GC-MS for all leaves and roots, all samples were previously treated with the methods of Hou et al. [20]. The metabolites of all samples were determined with gas chromatography (7890, Agilent) coupled with a time-of-flight mass spectrometer (Pegasus HT, LECO) by the method of Hou et al. [20]. The analysis of raw data was carried out by the method of Dunn et al. [21].

### 2.5. The Amino Acids Content of Leaves and Roots in E+ and E− Achnatherum inebrians

Total of 0.1 g dry powder of leaves and roots of E+ and E− *Achnatherum inebrians* was placed into 2 mL centrifuge tubes (Eppendorf), and extracted with 1 mL HCl (0.5 M), and vortexed for 20 min at 8000 rpm. After sonication with ultrasonic for 20 min at 25 °C, they were centrifuged for 20 min at 14,000× *g*. Finally, 250 µL supernatant was transferred to the liquid chromatographic sample bottle (Agilent ValueLab, 2 mL) added with ISTD, and 750 μL 80% acetonitrile aqueous solution (LC/MS) was added to each sample bottle. Hydrophilic interaction chromatography (HILIC) was used to separate the amino acids. The amino acid levels were determined by the comparison of retention times and peaks with corresponding standards (Sigma-Aldrich, St. Louis, MO, USA).

### 2.6. Calculation of Phosphorus Utilization Efficiency (PUE) of E+ and E− Achnatherum inebrians Plants

The calculation formula of PUE of *Achnatherum inebrians* is determined by the methods of Guo et al. [22], as follows: PUE*_leaf_* = $\frac{(W_{leaf})2}{P_{leaf}}$, where *W_leaf_* stands for the dry weight of leaves and *P_leaf_* stands for the P concentration in leaves; PUE*_root_* = $\frac{(W_{root})2}{P_{root}}$, where *W_root_* denotes the dry weight of roots and *P_root_* denotes the P concentration in roots; PUE*_total_* = $\frac{(W_{total})2}{P_{total}}$, where *W_total_* means the dry weight of whole plant and *P_total_* denotes the P concentration in whole plant.

### 2.7. Statistical Analysis

The difference of dry weight, the content of C, N, P, and metal ion, and phosphorus utilization efficiency in leaves and roots between E+ and E− *Achnatherum inebrians* was carried out with independent T-tests at 95% confidence level, and data were the means ± SE of six independent replicates. SIMCA-P 14.0 was used to carry out the supervised orthogonal projections to latent structures-discriminant analysis (OPLS-DA) and unsupervised principal component analysis (PCA). The ellipse indicated that the PCA score plots of the models define the 95% confidence interval of the modeled variations. VIP is the weighted sum of the squares of the OPLS-DA. Additionally, significantly different metabolites were identified with VIP > 1 and *p* < 0.05 (Student’s *t*-test). Subsequently, the metabolic pathways were calculated by MetaboAnalyst (http://metaboanalyst.ca/ accessed on 12 May 2021). PCA and heatmap of amino acids were performed by R, and amino acids content was determined by three independent replicates.

## 3. Results

### 3.1. Effects of Epichloë gansuensis on the Dry Weight of Achnatherum inebrians Seedlings at Various P Concentration

In our study, E+ and E− *Achnatherum inebrians* seedlings were treated with different P concentrations (0.01 mM and 0.5 mM) for 18 weeks. With the decrease of P concentration, the dry weight of leaves and roots of E+ and E− seedlings decreased significantly, and the leaves dry weight of E+ and E− seedlings at 0.01 mM P was two-fold and one-fold lower than that at 0.5 mM P, respectively (Figure 1a). Similarly, compared with 0.5 mM P, the root dry weight of E+ and E− seedlings decreased by one-fold and three-fold, respectively, at 0.01 mM P (Figure 1b). Additionally, we found that *Epichloë gansuensis* significantly increases the dry weight of leaves compared with the E− leaves at 0.01 mM P, and the leaves dry weight of E+ seedling was increased by 30.5% than that of E− seedling at the low P stress. However, there is no significant difference in the roots dry weight of E+ and E− seedlings at the low P stress. We also found that the dry weight of leaves and roots between E+ and E− seedlings at 0.5 mM P did not obviously differ, but under normal phosphorus supply conditions (0.5 mM), the dry weight of E+ leaves and roots was lower than E− (Figure 1a,b).

### 3.2. Effects of Epichloë gansuensis on C, N, and P Content of Achnatherum inebrians at Various P Concentration

We found that *Epichloë gansuensis* significantly increased the C content of roots at 0.01 and 0.5 mM P concentrations, and the C content of E+ roots at 0.01 and 0.5 mM P concentration was 1.55 and 1.14 times higher than that of E−, respectively (Figure 2d). However, there was no significant difference in C content of leaves between E+ and E− seedlings at 0.01 and 0.5 mM P concentrations (Figure 2a). Under 0.01 and 0.5 mM P concentrations, the *Epichloë gansuensis* did not cause significant changes in the contents of N in leaves and roots of seedlings (Figure 2b,e). However, at 0.01 mM P concentration, the N content of E+ and E− roots of seedling dramatically increased by 1.64 and 1.37 times respectively compared with that of 0.5 mM P concentration (Figure 2e). Additionally, we found that the N content of E+ leaves at 0.01 mM P concentration dramatically increased by 1.27 times compared with that of 0.5 mM P concentration (Figure 2b). *Epichloë gansuensis* did not cause significant changes in P contents in leaves and roots of seedlings under 0.01 and 0.5 mM P concentrations (Figure 2c,f). However, with the increase of P concentration, the P content in leaves and roots of E+ and E− seedlings significantly increased, and the P content of E+ and E− leaves at 0.01 mM P concentration was 1.03 times and 1.04 times lower than that at 0.5 mM P concentration, respectively. Moreover, the P content in E+ and E− roots at 0.01 mM P concentration was 1.1 times and 1.08 times lower than that at 0.5 mM P concentration, respectively (Figure 2f).

### 3.3. Effects of Epichloë gansuensis on the K, Ca, Na, Mg and Fe Content of Achnatherum inebrians at Low P Stress

We found that the K content of E+ and E− leaves was significantly different at 0.01 mM P and 0.5 mM P, and the K content of E+ and E− leaves increased by 1.15 and 1.3 times at 0.01 mM P compared with 0.5 mM P, respectively (Figure 3a). However, the K content of E+ and E− roots was significantly different only at 0.5 mM P, and the K content of E+ root was 1.6 times higher than that of E− (Figure 3d). In addition, a marked increase in the K content of leaves was observed in E+ and E− seedlings as P decreased, and the K content of E+ and E− leaves at 0.01 mM P was 1.17 and 1.3 times higher than that at 0.5 mM P, respectively. Similarly, the K content of roots of E− seedling was also significantly different under different P concentration, and the K content of E− roots at 0.01 mM P was 1.57 times lower than that at 0.5 mM P. The Ca content in leaves and roots of E+ seedlings was 2.3 times higher and 1.71 times lower than that of E− seedlings, respectively, at 0.5 mM P (Figure 3b,e). The Ca content of roots of E+ seedlings was 1.16 times lower than that of E− seedlings in 0.01 mM P (Figure 3e). The Ca content of E− leaves and roots increased by 1.86 times and decreased by 1.85 times at 0.01 mM P compared with 0.5 mM P, respectively. Na content of E+ leaves and roots decreased by 1.52 times and increased by 1.45 times than that of E− seedlings, respectively (Figure 3c,f). Na content of E− leaves and roots at 0.01 mM P decreased by 1.4 times and increased by 1.55 times compared with 0.5 mM P, respectively. Mg content of leaves of E+ seedlings and E− seedlings was increased by 1.23 and 2.94 times in 0.5 mM P compared with 0.01 mM P, respectively (Figure 4a). Mg content of E+ and E− leaves at 0.01 mM P decreased by 1.12 times and 2.7 times compared with 0.5 mM P. However, the Mg content of E+ and E− seedling roots was significantly different only at 0.5 mM P, and Mg content of E+ roots was 1.84 times higher than that of E− (Figure 4c). The Fe content in leaves and roots of E− seedlings was significantly different under the different P, and the Fe contents in leaves and roots of E− at 0.01 mM P decreased by 4.11 times and increased by 3.2 times compared with 0.5 mM P, respectively (Figure 4b,d). However, there was no significant change of Fe content in leaves and roots of E+ seedlings under the different P concentrations. In addition, under 0.01 mM P concentration, the Fe content in the leaves of E+ seedlings was significantly lower than that of E− seedlings, and the Fe content of E+ leaves was 1.98 times less than that of E−.

### 3.4. Metabolic Changes

In our experiment, 30 and 22 different metabolites of E+ and E− seedlings were found at 0.01 mM P and 0.5 mM P, respectively. Further, according to the score plot of PCA, there was a marked separation among the four groups of samples in the leaves (E+ leaves at 0.01 mM P: 0.01LE+; E− leaves at 0.01 mM P: 0.01LE−; E+ leaves at 0.5 mM P: 0.5LE+; E− leaves at 0.5 mM P: 0.5LE−). As shown in the Figure 4a, PC1 and PC2 account for 29.2% and 13.1% of the total variance, respectively. The data of metabolomics were performed out with OPLS-DA method to estimate that obvious differences in the metabolic profiles of E+ and E− *Achnatherum inebrians* at 0.01 and 0.5 mM P, respectively, were found. We carried out pairwise comparison of the data obtained by using OPLS-DA, and the result exhibited a significant distinction between 0.01LE− and 0.01LE+ (R2Y = 0.986; Q2 = 0.559) (Figure 4b); between 0.5LE− and 0.5LE+ (R2Y = 0.974; Q2 = 0.142) (Figure 4c); between 0.01LE+ and 0.5LE+ (R2Y = 0.998; Q2 = 0.947) (Figure 4d); between 0.01LE− and 7.5LE− (R2Y = 0.997; Q2 = 0.920) (Figure 4e). They have good model quality that showed under 0.01 mM P, the leaves of E+ and E− plants are different from those of 0.5 mM P, respectively (Figure 4d,e), it showed that metabolic profiles were in a P concentration-dependent manner. There was clear separation between the leaves of E+ and E− plants, respectively, under 0.01 and 0.5 mM P, (Figure 4b,c), and metabolic profiles were in an *Epichloë gansuensis*-dependent manner. According to the dramatic importance of *T*-test (*p* < 0.05) and VIP (VIP > 1) score, the metabolites with clear differences were selected. So, we speculated *Epichloë gansuensis* probably regulated the metabolism pathway in the leaves of *Achnatherum inebrians* to adapt to different P concentrations.

Similarly, the results of PCA found that there was an obvious separation among the four groups of samples in the roots (E+ roots at 0.01 mM P: 0.01RE+; E− roots at 0.01 mM P: 0.01RE−; E+ roots at 0.5 mM P: 0.5RE+; E− roots at 0.5 mM P: 0.5RE−). PC1 and PC2 account for 43.2% and 7.94% of the total variance in the roots, respectively (Figure 5a). Our results found that there was a clear distinction between 0.01RE+ and 0.01RE− (R2Y = 1; Q2 = 0.530) (Figure 5b); between 0.5RE+ and 0.5RE− (R2Y = 1; Q2 = 0.341) (Figure 5c); between 0.01RE+ and 0.5RE+ (R2Y = 0.999; Q2 = 0.947) (Figure 5d); between 0.01RE− and 0.5RE− (R2Y = 0.998; Q2 = 0.940) (Figure 5e), and which showed they have good model quality; and the metabolic profiles in the roots of E+ and E− plants at 0.01 mM P are markedly different from those of 0.5 mM P (Figure 5d,e). Generally, metabolic changes were in a P-dependent manner in the roots. There was significant difference between the roots of E+ and E− seedlings, respectively, at 0.01 and 0.5 mM P (Figure 5b,c). Generally, metabolic changes occurred in an *Epichloë gansuensis*-dependent manner in the roots. The metabolites with clear difference (VIP > 1, *p* < 0.05) were selected. So, we inferred the *Epichloë gansuensis* probably reprograms the metabolism pathway in *Achnatherum inebrians* roots to adapt to the different P concentrations.

### 3.5. Effects of Epichloë gansuensis on Metabolic Profiles in the Leaves of Achnatherum inebrians at 0.01 and 0.5 mM P

We found that there were 13 and 4 metabolites that significantly change between LE+ (the leaves of E+ plants) and LE− (the leaves of E− plants) at 0.01 mM and 0.5 mM P, respectively (Appendix A). The results showed that *Epichloë gansuensis* decreased the content of histidine (8.23 × 10-8-fold), lactulose (0.04-fold), citric acid (0.48-fold), 1-Kestose (0.09-fold), sedoheptulose (0.19-fold), 4-hydroxycinnamic acid (0.18-fold), 1-hexadecanol (0.30-fold), ascorbate (0.29-fold) and pyruvic acid (0.60-fold), and increased the contents of 5-aminovaleric acid (3.70-fold), vanillin (5.90-fold), D-glutamic acid (3.16-fold), and 4-acetamidobutyric acid (2.90-fold) in LE+ at 0.01 mM P compared to LE− (Appendix A). The result also showed that *Epichloë gansuensis* decreased the contents of benzoic acid (0.18-fold) and cortexolone (0.17-fold), and increased the content of citramalic acid (1.80-fold) and salicylic acid (2.83-fold) in LE+ compared to LE− at 0.5 mM P (Appendix A). However, our results found that there were no significantly different metabolites of LE+ vs. LE− between 0.01 and 0.5 mM P (Figure 6a).

There were 31 and 32 clearly different metabolites between 0.01 and 0.5 mM P in LE+ and LE−, respectively (Appendix A). Low P stress decreased the content of glucose-6-phosphate (0.02-fold), citric acid (0.42-fold), 1-kestose (0.02-fold), and phosphomycin (0.27-fold) in LE+ compared with 0.5 mM P, and increased the content of lysine (3131237-fold), glutamine (2417309-fold), biuret (412586-fold), D-glutamic acid (315215-fold), asparagine (10195.77-fold), noradrenaline (108640.5-fold), orotic acid (49037.5-fold), O-acetylserine (12707-fold), saccharic acid (6.93-fold), vanillin (98776.5-fold), benzoic acid (5.21-fold), urea (324.46-fold), 3-cyano-L-alanine (492.24-fold), glycocyamine (8.72-fold), taxifolin (76824.5-fold), alpha-ketoisocaproic acid (19638-fold), xylitol (4.21-fold), 4-methylcatechol (5233.5-fold), thymol (4.10-fold), tyrosine (40.56-fold), proline (31.93-fold), thymidine (9.02-fold), indole-3-acetamide (38.31-fold), isoleucine (12.34-fold), putrescine (8.70-fold), itaconic acid (2.00-fold), and phenylalanine (8.10-fold) in LE+ compared with 0.5 mM P (Appendix A). Low P stress decreased the content of alpha-tocopherol (0.09-fold), 1-aminocyclopropanecarboxylic acid (0.09-fold), glucose-6-phosphate (0.05-fold), gentisic acid (0.12-fold), pelargonic acid (0.17-fold), 6-phosphogluconic acid (0.11-fold), and maltotriose (0.18-fold) in LE− compared with 0.5 mM P, and increased the content of asparagine (32088.80-fold), lysine (2496491.5-fold), glutamine (1790354-fold), 4-acetamidobutyric acid (1893323.5-fold), biuret (346730.5-fold), noradrenaline (137939.5-fold), 3-cyano-L-alanine (516.31-fold), histidine (60689.5-fold), D-glutamic acid (99723.5-fold), urea (183.76-fold), thymidine (21.65-fold), putrescine (16.00-fold), oxalacetic acid (5.21-fold), tyrosine (88.76-fold), orotic acid (8.80-fold), o-acetylserine (12386-fold), itaconic acid (4.17-fold), glycocyamine (14.48-fold), proline (20.13-fold), phthalic acid (215.51-fold), indole-3-acetamide (8.62-fold), cholic acid (13.40-fold), pantothenic acid (4.84-fold), isoleucine (10.11-fold), and phenylalanine (6.66-fold) in LE− compared with 0.5 mM P (Appendix A). Interestingly, our results found that there were 20 same clearly different metabolites in 0.01 mM P vs. 0.5 mM P between LE+ and LE− by Venn diagram analysis, including glucose-6-phosphate, lysine, glutamine, biuret, D-glutamic acid, asparagine, noradrenaline, orotic acid, o-acetylserine, urea, 3-cyano-L-alanine, glycocyamine, tyrosine, proline, thymidine, indole-3-acetamide, isoleucine, putrescine, itaconic acid, and phenylalanine (Figure 6b).

### 3.6. Effects of Epichloë gansuensis on Metabolic Profiles in the Roots of Achnatherum inebrians at 0.01 and 0.5 mM P

We found that there were 17 and 18 metabolites that changed dramatically between RE+ and RE− at 0.01 and 0.5 mM P, respectively (Appendix A). Our results showed that *Epichloë gansuensis* decreased the content of ribonic acid (0.17-fold), tryptophan (0.22-fold), maleic acid (0.50-fold), cortexolone (0.41-fold), 5-aminovaleric acid (0.55-fold), proline (0.43-fold), acetol (0.58-fold), leucine (0.43-fold), maleamate (0.52-fold), ethanolamine (0.57-fold), and alanine (0.60-fold), and increased the contents of *cis*-1,2-dihydronaphthalene-1,2-diol (17.42-fold), gluconic acid (5.62-fold), 3-methylglutaric acid (1.81-fold), asparagine (1.87-fold), D-glutamic acid (1.89-fold), and 3-cyano-L-alanin (1.68-fold) in RE+ at 0.01 mM P concentration compared to RE− (Appendix A). The result also showed that *Epichloë gansuensis* only decreased the content of 1,3-diaminopropane (0.48-fold) and increased the content of galactonic acid (2052.00-fold), 1-kestose (6.17-fold), uridine (28.43-fold), isomaltose (6.03-fold), asparagine (5.81-fold), *cis*-gondoic acid (3.77-fold), phosphate (2.57-fold), 3-hydroxybutyric acid (2.82-fold), 3-cyano-L-alanine (2.74-fold), proline (2.03-fold), 1-hexadecanol (1.98-fold), aspartic acid (1.89-fold), alanine (1.64-fold), gluconic acid (1.79-fold), tyrosine (1.61-fold), maltotriose (1.67-fold), and glutamic acid (1.55-fold) in RE+ compared to RE− at 0.5 mM P (Appendix A). Further, between 0.01 and 0.5 mM P, RE+ vs. RE− have five same markedly different metabolites, including gluconic acid, proline, asparagine, alanine, and 3-cyano-L-alanine (Figure 7a).

We also found that there were 24 and 31 differentially expressed metabolites between 0.01 and 0.5 mM P in RE+ and RE−, respectively (Appendix A). Low P stress caused negative effects on the content of 6-phosphogluconic (0.05-fold), 1-aminocyclopropanecarboxylic acid (0.21-fold), glutaric acid (0.25-fold), glucose-6-phosphate (8.3 × 10^−5^-fold), phosphate (0.35-fold), 1-kestose (0.11-fold), uridine (0.03-fold), phosphomycin (0.27-fold), lauric acid (0.43-fold), threonine (0.13-fold), and fructose (0.40-fold) in RE+ compared to 0.5 mM P, and the 0.01 mM P had positive effect on the content of biuret (21.89-fold), glutamine (104454.43-fold), D-glutamic acid (145640.00-fold), benzoic acid (44405.00-fold), urea (197.28-fold), lysine (12.59-fold), asparagine (243.35-fold), phthalic acid (95.58-fold), 3-cyano-L-alanine (2.85-fold), ribitol (4.11-fold), aspartic acid (6.62-fold), L-allothreonine (4.69-fold), and glycine (4.06-fold) in RE+ compared to the 0.5 mM P. Additionally, low P stress decreased the content of *cis*-1,2-dihydronaphthalene-1,2-diol (0.04-fold), 6-phosphogluconic acid (5.5 × 10^−5^-fold), fructose (0.28-fold), lauric acid (0.23-fold), threonine (0.07-fold), glutaric acid (0.35-fold), and threitol (0.42-fold) in RE− compared to the 0.5 mM P, and the 0.01 mM P increased the content of glutamine (1494750.67-fold), biuret (701374.00-fold), d-glutamic acid (77177.67-fold), lysine (34.04-fold), benzoic acid (5.71-fold), gluconic lactone (21647.33-fold), asparagine (857.04-fold), glycocyamine (8747.00-fold), urea (40.74-fold), maleic acid (5.31-fold), ribitol (4.11-fold), 3-cyano-L-alanine (66.10-fold), phthalic acid (27.13-fold), tryptophan (4.24-fold), aspartic acid (10.94-fold), proline (8.93-fold), alanine (7.45-fold), glycine (7.05-fold), oxoproline (5.50-fold), L-allothreonine (5.24-fold), maleamate (5.02-fold), serine (4.82-fold), glutamic acid (4.49-fold), and thymidinein (3.42-fold) in RE− compared to the 0.5 mM P. Significantly, there were 18 same significant different metabolites of 0.01 mM P vs. 0.5 mM P between LE+ and LE− by Venn diagram analysis, including asparagine, glutamine, d-glutamic acid, benzoic acid, 6-phosphogluconic acid, urea, biuret, lysine, glutaric acid, phthalic acid, 3-cyano-L-alanine, ribitol, lauric acid, threonine, aspartic acid, L-allothreonine, fructose, and glycine (Figure 7b).

### 3.7. Effects of Epichloë gansuensis on Metabolic Pathways in the Leaves of Achnatherum inebrians at 0.01 and 0.5 mM P

We estimated the impact value by pathway topology analysis and *p* value by enrichment analysis to research the differences in metabolic pathways between LE+ and LE− at 0.01 and 0.5 mM P, respectively. We found that *Epichloë gansuensis* most probably regulated metabolic pathways, including biosynthesis of secondary metabolites; ascorbate and aldarate metabolism; pyruvate metabolism; valine, leucine and isoleucine biosynthesis in leaves of *Achnatherum inebrians* at 0.01 mM P (Figure 8a), and however, there were no correlated pathways differentially regulated by *Epichloë gansuensis* in the leaves at 0.5 mM P. This indicated that *Epichloë gansuensis* plays an important role in reprogramming the metabolism of host leaves to adapt to low P stress. We also found that phenylalanine metabolism, isoquinoline alkaloid biosynthesis, cyanoamino acid metabolism, alanine, aspartate, and glutamate metabolism in LE+ may be the most correlative pathways regulated by low P stress (Figure 8b). Similarly, phenylalanine metabolism, isoquinoline alkaloid biosynthesis, cyanoamino acid metabolism, alanine, aspartate, and glutamate metabolism in the LE− maybe the most relevant pathways differentially influenced by low P stress (Figure 8c). Therefore, *Epichloë gansuensis* probably reprogrammed the amino acids metabolism of leaves to adapt the different P concentrations.

### 3.8. Effects of Epichloë gansuensis on Metabolic Pathways in the Roots of Achnatherum inebrians at 0.01 and 0.5 mM P

We found that cyanoamino acid metabolism; tyrosine metabolism; arginine and proline metabolism; tryptophan metabolism in the roots may be the most related pathways regulated by *Epichloë gansuensis* at 0.01 mM P (Figure 9a), and the alanine, aspartate, and glutamate metabolism; cyanoamino acid metabolism; isoquinoline alkaloid biosynthesis; arginine and proline metabolism may be the most relevant pathways regulated by *Epichloë gansuensis* in the roots at 0.5 mM P (Figure 9b). We can speculate that *Epichloë gansuensis* has a significant role in regulating the amino acids metabolism of the roots in *Achnatherum inebrians* to adapt to different P concentrations. Furthermore, the low P stress most probably influenced the pathway of cyanoamino acid metabolism; glycine, serine, and threonine metabolism; alanine, aspartate, and glutamate metabolism in RE+ (Figure 9c), the low P stress most probably regulated the pathway of alanine, aspartate, and glutamate metabolism; glycine, serine, and threonine metabolism; cyanoamino acid metabolism in the RE− (Figure 9d). Interestingly, we found that *Epichloë gansuensis* influenced the metabolic pathway of cyanoamino acid in roots at 0.01 mM P and 0.5 mM P, which indicated this pathway was important for low P stress, instead of *Epichloë gansuensis*. In summary, *Epichloë gansuensis* probably regulated the amino acids metabolism of roots to adapt the different P concentrations.

### 3.9. The Changes of Amino Acids in Leaves and Roots of E+ and E− Achnatherum inebrians at Low P Stress

Some variables displayed significant separation, and the factor 1 explained 60.66% of the total variance for amino acid in leaves, of which glutamine content contributed the most variation (28.5%), while tryptophan content and aspartic acid content accounted for approximately 27.9% and 27.6%, respectively (Figure 6a). *L*-isoleucine content (55.6%) and tyrosine content (42.3%) were loaded on factor 2, which could explain 12.42% of the variation for amino acid in leaves. The significant differences between 0.01 and 0.5 mM P from the leaves of E+ and E− plants were found in Figure 10a. In addition, endophyte infection could explain 0.6% variation for amino acids of leaves (Adonis, *p* = 1) (Figure 10a); P-treatment accounted for 75.7% variation for amino acids of leaves (Adonis, *p* = 0.001) (Figure 10a); the interaction of P-treatment × endophyte-infection accounted for 15.2% variation for amino acids of leaves (Adonis, *p* = 0.005) (Figure 10a). Similarly, the factor 1 explained 71.03% of the total variance for amino acids in roots, of which glutamine content and lysine content contributed the most variation (26.7% and 26.7%, respectively), while leucine content accounted for approximately 26.6% (Figure 10b). Threonine content (51.5%) and tyrosine content (32.4%) were loaded on factor 2, which could explain 15.39% of the variation for amino acids in roots. In addition, endophyte infection could explain 4.3% variation for amino acids of roots (Adonis, *p* = 0.003) (Figure 10b); P-treatment accounted for 90.6% variation for amino acids of roots (Adonis, *p* = 0.001) (Figure 10b); the interaction of P-treatment × endophyte-infection accounted for 2.1% variation for amino acids of roots (Adonis, *p* = 0.063) (Figure 10b). Further, our results showed that with the decrease of P concentration, tyrosine, histidine, *L*-isoleucine, DL-methionine, *L*-asparagine in E− leaves were up-regulated, while some amino acids were down regulated by low P stress, mainly including threonine and alanine. Interestingly, *Epichloë gansuensis* up-regulated the content of 14 amino acids in leaves, including threonine, alanine, serine, L-asparagine, proline, glycine, glutamic acid, tryptophan, aspartic acid, lysine, phenylalanine, glutamine, and histidine at 0.01 mM P, meanwhile, *Epichloë gansuensis* decreased the content of leucine, valine, *L*-isoleucine, and DL-methionine at 0.01 mM P (Figure 10c). Low P stress led to many changes in the accumulation of amino acids in the root—low P stress increased almost all amino acids, including threonine, proline, tryptophan, tyrosine, leucine, L-isoleucine, alanine, phenylalanine, histidine, glutamine, and lysine, which were significantly up-regulated (Figure 10d). However, *Epichloë gansuensis* up-regulated seven kinds of amino acids in roots under low P stress, for example glycine, serine, aspartic acid, arginine, *L*-asparagine, valine, and glutamic acid; at the same time, the levels of threonine, proline, tryptophan, tyrosine were down regulated (Figure 10d).

### 3.10. Effects of Epichloë gansuensis on Phosphorus Utilization Efficiency (PUE) of Achnatherum inebrians at Low P Stress

As shown in the Figure 11, regardless of the presence of *Epichloë gansuensis*, the low P stress decreased the phosphorus use efficiency (PUE) of leaves, roots, and total in E+ and E− *Achnatherum inebrians* plants, and the leaves PUE, roots PUE, and total PUE at low P stress were decreased by 44.9%, 56.6%, and 51.8% compared with 0.5 mM P in the E+ plants, respectively (Figure 11a–c). The leaves PUE, roots PUE, and total PUE at low P stress were decreased by 70.1%, 73.5%, and 72.2% compared with 0.5 mM P in the E− plants, respectively (Figure 11a–c). Moreover, *Epichloë gansuensis* increased leaves PUE and total PUE at low P stress, and the leaf PUE and total PUE of E+ plants were increased by 68.5% and 45.9% compared with E− plants at 0.01 mM P, respectively (Figure 11a,c).

## 4. Discussion

The *Epichloë* endophyte exists in many cool-season grasses, including *Achnatherum inebrians*, *Festuca arundinacea*, *Hordeum brevisubulatum*, and *Lolium perenne* [11,12,15,23]. The relationship between *Epichloë* endophyte and host grasses is mutually beneficial as has been confirmed by previous studies [24]. The present study has indicated that the presence of the *Epichloë gansuensis* in *Achnatherum inebrians* plants enhanced the dry weight of leaves, leaves PUE, and total PUE, amino acids content, and reprogrammed the amino acids metabolic process and regulated metal ion contents when plants were growing at the low P stress.

Phosphorus is an essential macro-element for the plant life cycle. It is the main factor limiting the increase of crop yield. The low P in soil can limit plant growth and development [25]. Our results showed that the low P stress significantly reduced the dry weight of leaves and roots in *Achnatherum inebrians* compared with 0.5 mM P, however, *Epichloë gansuensis* could increase dry weight at 0.01 mM P, which indicated that *Epichloë gansuensis* could improve the tolerance of *Achnatherum inebrians* seedlings to phosphorus deficiency stress. One study was similar to our results, and it had shown that *Epichloë* endophyte can improve the absorption of phosphorus by changing the root morphology and increasing root exudates, thus enhancing the fresh and dry weight of the host plant leaves and roots at low P conditions (5 mg kg^−1^ soil) [26]. Previous studies have shown that the effect of *Neotyphodium coenophialum*-endophyte on the growth of tall fescue depends on soil P content, and show a more active role in high P conditions (25 mg kg^−1^ soil) compared with normal P supply level (5 mg kg^−1^ soil) [26]. In the present study, dry weight of E+ plants decreased under 0.5 mM P and endophytes may lead to a certain cost of host plant growth, such as competing with host plants for nutrients and photosynthetic products. Different treatment times, host plant genotype, and endophyte species all have an impact on the research results. Chen et al. found that the dry weight of E+ wild barley was lower than that of E− without NaCl stress [27]. Therefore, *Epichloë* endophyte mainly increases dry weight by improving the photosynthetic and root morphology to adapt to low phosphorus stress [28], but not sensitive to the soil supplied with sufficient phosphorus.

One study reported the effects of *Epichloë* endophyte infection on nitrogen utilization [10], but there are fewer studies on low P stress. In the present experiment, *Epichloë gansuensis* affected the nutrient absorption and utilization of *Achnatherum inebrians* at different P levels. The change of soil P concentrations directly affects the absorption and accumulation of P by plants. Our research found that the P content in E+ and E− tissues of *Achnatherum inebrians* reduced clearly with the decrease of P level. However, *Epichloë gansuensis* did not significantly affect the P content in leaves and roots at different P concentrations, which was contrary to previous studies [15]. It has been confirmed that many microorganisms play an important role in enhancing the P absorption of the host at P deficiency conditions [29]. This might be due to the typical structures of hyphae and micro sclerotia in root cells by endophyte associated with roots [30]. The opposite result was probably caused by different growth stages, different time and concentrations of P treatment, and different genotypes of endophyte and plant combinations.

P deficiency stress can interfere with the absorption of other nutrients [15]. We found that the N content of leaves and roots of *Achnatherum inebrians* at 0.01 mM P level were increased compared with 0.5 mM P. One study reported that P deficiency can affect the activity of nitrate reductase (NR) in *Rosa roxburghii*, resulting in the decrease of N content [31]. Therefore, we speculated that *Epichloë gansuensis* plays an important role in this process. However, low P had no significant effect on C content in leaves and roots of E+ and E− plants, which may be due to that the low P stress has limited C uptake by *Epichloë gansuensis* and plants. Interestingly, *Epichloë gansuensis* increased the content of C in roots, but had no significant effect on leaves, which implied that *Epichloë gansuensis* may control the transfer of host C nutrients. This is similar to a previous study that suggested that the effect of endophyte on the nutrient content of plant roots is greater than that of aboveground parts [32]. On the other hand, it was reported that PGPR (plant growth promoting rhizobacteria) can affect root structure and promote plant growth [33]. In turn, plants will release more C in root exudates to increase microbial activity, and maintain microbial function. In addition, other studies have reported that the increase of plant nutrient uptake is related to the increase of root surface area. Perhaps *Epichloë gansuensis* enhances root exudation and microbial activity, thus leading to more nutrients for plant roots [34].

In the process of P absorption, plants can help or antagonize the absorption of other nutrients. Potassium (K) is the only monovalent cation among the essential elements of plants, and it is also an activator of enzymes in many important physiological processes, such as protein synthesis, sugar transport, N and C metabolism, and photosynthesis, which is very important to ensure the optimal growth of plants [35]. We found that K content in the leaves of *Achnatherum inebrians* was significantly higher at low P stress than that at 0.5 mM P, and *Epichloë gansuensis* also remarkably increased the K content in leaves. The results were consistent with the previous study [36]. They reported that the K content in roots of *Aleurites montana* was higher than that in leaves under normal P supply, but lower than that in leaves under P deficiency, which might be due to the fact that the content of mineral elements in leaves is directly proportional to the level of photosynthesis, and endophyte can promote the photosynthesis of host plants. *Epichloë gansuensis* could significantly increase the contents of Ca in leaves and Na in roots at 0.5 mM P, and the content of Mg and Fe in E+ leaves was significantly lower than that in E+ at 0.01 mM P. Research has shown that there was a competitive mechanism between Ca and Mg [37]. Interestingly, it seemed that *Epichloë gansuensis* did not significantly affect the content of Ca, Na, Mg, and Fe in different tissues of *Achnatherum inebrians* at low P stress. However, the content of Na, Mg, and Fe in the root was significantly higher than that in the leaves at 0.01 mM P, which indicated that the plant itself might change the metal accumulation in leaves by chelating iron in roots under nutrient stress, thus plants can adapt to soil acidity associated with high levels of exchangeable Fe [14]. In the process of soil environment and plant metabolism, P can inhibit the absorption and transfer of iron in plants, and due to the inhibition of P deficiency on iron transfer mechanism, iron accumulation in roots increased under both phosphorus deficiency and excess phosphorus levels [38].

Adjusting metabolic pathways is an important adaptive response to low P stress [39]; in these processes, amino acids are the main players in various metabolic and regulatory pathways. Threonine and methionine are precursors of isoleucine synthesis, so the down-regulation of threonine may be to ensure the balance between threonine and methionine at the abiotic stress [40]. *Epichloë gansuensis* increased the biosynthesis of aminoacyl-tRNA in leaves by increasing the content of alanine and serine, meanwhile, decreasing the content of valine. One study has shown that phenylalanine, proline, and threonine play important roles in signal transduction and stress response [41]. Therefore, endophyte improves N metabolism, C metabolism and photosynthesis. Similar to the amino acid metabolism of leaves, the amino acid levels of roots was almost all up-regulated at low P stress; this may be due to the close contact between root and soil P. *Epichloë gansuensis* induced up-regulation of arginine in roots. Arginine is an important amino acid for nitrogen transport and storage, and also a precursor for the synthesis of other amino acids [42]. However, contrary to the situation in leaves, *Epichloë gansuensis* down regulated the levels of threonine, tryptophan, and tyrosine. *Epichloë gansuensis* may change the general reaction pattern of most amino acids in roots at low P stress, which promotes protein synthesis.

Metabolomics is a good method to analyze various small molecular metabolites in plants in response to environmental stresses. The change of metabolites may truly reflect the integration of genomic regulation and protein expression under the environment stresses. In this study, we explored the changes of metabolites in leaves and roots of *Achnatherum inebrians* under low P stress. Our results showed that *Epichloë gansuensis* increased the tolerance of *Achnatherum inebrians* to low P stress through the regulation of different metabolites, metabolic pathway, and amino acids content. One study has shown that plants in low P environment can change their carbon metabolism, including the secretion of some low molecular weight organic acids and enzymes [2]. The C metabolism mainly includes sugar metabolism, organic acid metabolism, and metabolism related to photosynthetic enzymes [43]. Our study showed that although low phosphorus stress did not significantly affect the carbon content of leaves and roots, it changed the carbon metabolism, including the metabolite content of some sugars and organic acids (Appendix A) and amino acid metabolic pathway (Figure 8 and Figure 9). In the present study, *Epichloë gansuensis* increased the content of vanillin in leaves at low P stress, and vanillin can increase the tolerance of plants to oxidative stress by scavenging reactive oxygen species [44], which indicates that *Epichloë gansuensis* induces resistance and raise the expression of stress-related metabolites. In addition, *Epichloë gansuensis* reduced pyruvate content compared with E− leaves in the TCA cycle at low P stress, thus E+ plants accumulate more energy to resist low P stress. Secretion of organic acids by plants at normal nutrition was less compared to low nutrition conditions, so as to promote the absorption of nutrients and improve the ability of plants to adapt to stress [45], The research demonstrated that *Epichloë gansuensis* increased the contents of 5-aminovaleric acid and 4-acetamidobutyric acid in leaves compared to LE− at low P stress. As a precursor of 4-aminobutyric acid (GABA), 4-acetamidobutyric acid is a marker of stress response and plays an important role in N and C metabolism [46], and each pathway participates in the shunt of GABA in TCA cycle, and TCA cycle was an important metabolic pathway [47]. Citric acid, as a basic metabolite of plants, participates in energy metabolism and substance transformation, and accumulates massively under the abiotic stress. However, our results showed that the content of citric acid and ascorbate in leaves was decreased by *Epichloë gansuensis* at low P stress, which may be related to the degree and time of low P stress, and species sensitivity. Our results also found that the *Epichloë gansuensis* mainly affected the content of various metabolites in the roots at low P stress. For example, amino acid (tryptophan; proline; leucine; D-glutamic acid; alanine; asparagine), organic acid (ribonic acid; gluconic acid), alcohols (*Cis*-1,2-dihydronaphthalene-1,2-diol). *Cris*-1,2-dihydronaphthalene-1,2-diol participate in the TCA cycle [48], and the accumulation of its content indicates that *Epichloë gansuensis* promotes the TCA cycle; therefore, *Epichloë gansuensis* influenced C metabolism. A large number of studies have confirmed that asparagine is a nitrogen carrier, involved in the storage and transportation of nitrogen, used to mobilize nitrogen from the source to the sink [49]. We found that *Epichloë gansuensis* improved the accumulation of asparagine in the roots at low P stress, which may increase N content in the roots through nitrogen remobilization, which is consistent with our results; therefore, *Epichloë gansuensis* also influenced N metabolism. We observed a decrease in the levels of organic acids, such as ribonic acid, associated with the biosynthesis and metabolism of fatty acids in E+ roots at low P stress. Studies have shown that the release of fatty acids from cell membrane is related to stress tolerance [50], so *Epichloë gansuensis* may adapt to low P stress by reducing the metabolism of fatty acids. In addition, the accumulation of gluconic acid in E+ roots increased significantly under low P stress, indicating that *Epichloë gansuensis* may promote root growth and improve adaptability, which is consistent with the previous studies [51].

On the other hand, *Epichloë gansuensis* has different effects on the process of reprogramming metabolites at different P concentrations. *Epichloë gansuensis* changes the metabolism of host leaves at 0.5 mM P by mainly affecting the content of organic acids, including benzoic acid, citramalic acid, and salicylic acid. LE+ accumulated more content of salicylic acid and citramalic acid at 0.5 mM P, indicating that *Epichloë gansuensis* may prefer the environment with sufficient nutrition to increase salicylic acid and citramalic acid. Benzoic acid is a key regulator of the interaction between plant and environment, and plays an important role in plant adaptability to harsh environment [52]. *Epichloë gansuensis* reduce the content of benzoic acid by regulating the phenylalanine pathway, which may be the adaptive strategy of endophyte at the sufficient nutrients condition. We found the most direct effect was that *Epichloë gansuensis* increased phosphate content in the root at 0.5 mM P. The increase of amino acids and organic acids in root exudates changed the respiration rate, which affected the root micro symbiotic community and led to the change of light efficiency [53], *Epichloë gansuensis* increased protein synthesis by increasing the content of aspartic acid, alanine, tyrosine, and glutamic acid in roots, further influencing N nitrogen. In addition, *Epichloë gansuensis* promotes C metabolism by increasing the content of isomaltose and maltotriose. The changes of C metabolism caused by low P stress lead to different carbohydrate levels in the underground part, which helps to increase root biomass and change root morphology [54]. Therefore, *Epichloë gansuensis* mainly regulates root C metabolism and N metabolism at the 0.5 mM P.

We found that low P stress mainly influences the content of amino acids (lysine, glutamine, D-glutamic acid, asparagine, O-acetylserine, 3-cyano-L-alanine, glycocyamine, tyrosine, proline, isoleucine) in LE+, and low P stress regulates the metabolism of proline by pathway analysis. One study has shown that P deficiency may reduce the activity of aquaporins, thereby reducing whole plant water potential [55]. Therefore, the high accumulation of proline may explain our findings. The content of glutamine, tryptophan, tyrosine, proline, and phenylalanine is higher, and the accumulation in LE+ was helpful to adapt to low P stress [56]; for example, the accumulation of phenylalanine in maize leaves at low P environment, which is related to the synthesis of plant secondary metabolites [57]. More notably, glucose-6-phosphate was significantly reduced in leaves—the reduction of these phosphorylated metabolites has seriously affected many metabolic processes in plants [44]. Previous studies have shown that some polyamines, such as putrescine, play an important role as antioxidants in plant responses to various environmental stresses [58], which was consistent with our research. In RE+, the low P stress mainly increased the content of amino acids (asparagine, glutamine, D-glutamic acid, lysine, aspartic acid, and glycine) and organic acid (phthalic acid), instead, decreased the content of amino acids (threonine), organic acids (6-phosphogluconic acid), fatty acids (lauric acid), and sugar (fructose and glucose-6-phosphate). Glucose-6-phosphate and 6-phosphogluconic acid are the key intermediate metabolites in the pentose phosphate pathway (PPP), glucose 6-phosphate is oxidized by G6PDH to produce NADPH, and NADPH as reducing power to scavenge ROS [59]. It had been shown that plants maintain membrane fluidity through desaturation of fatty acid [60]. P deficiency leads to the increase of most amino acids and organic acids in roots, which is due to the inhibition of protein synthesis and the accumulation of a large number of amino acids [61], such as Asn and Gln in our study. P deficiency also greatly inhibited the phosphorylation of hexose in roots, including fructose.

Interestingly, low P stress mainly affected the metabolism of phenylalanine, tyrosine, and isoleucine in LE−. Isoleucine can form alkaline phosphatase, which is synthesized in large quantities in abiotic stress, and phenylalanine and tyrosine biosynthesis pathways were up-regulated, which indicated that the synthesis of aromatic amino acids was up-regulated, which played a key role in the synthesis of other secondary metabolites and the promotion of plant growth and development at abiotic stress [62]. Phenols have been described as the marker of the tolerance of abiotic stress in plants [63], but their content vary with environment, stress degree, and species sensitivity, which may explain the phenomenon that low P stress causes the decrease of tocopherol content in leaves in this study. However, low P stress mainly increased the content of amino acids in RE−.

The use of more nutritionally efficient crops is essential for improving environment adaptability and maintaining yield [64]. Effective utilization of P is considered to be an important adaptation for plant growth in low P soils. PUE refers to the ability of plant to produce higher dry matter per unit of P absorbed [65]. Our study found that PUE of leaves and roots was significantly lower at low P stress than that at 0.5 mM P. The reutilization of organic P is of great significance for maintaining plant growth and improving PUE [66]. Therefore, low P may limit the activation process of organic P in tissues. Moreover, the PUE of roots was higher than that of leaves, especially under low phosphorus stress. This may be related to the adaptation mechanism of plants to low phosphorus, with greater root biomass and higher root activity beneficial to the improvement of PUE. However, the colonization of endophyte can improve the growth in phosphorus-deficient environment by increasing the PUE, which is the same as a previous study [30]. Therefore, *Epichloë gansuensis* may improve the P status of *Achnatherum inebrians* by improving PUE rather than tissue phosphorus concentration in leaves and roots in different degrees. Xu et al. show that endophyte promoted the biomass, phosphorus absorption, and photosynthesis of inoculated maize by enhancing phosphorus enzyme activities in the rhizosphere and by decreasing the pH of the rhizosphere compared with non-inoculated controls, either in sufficient or deficient phosphorus conditions in pot cultures [30]. Besides, only under stress conditions (such as low phosphorus conditions) can endophyte help plants increase phosphorus utilization efficiency and regulate their own consumption and competition of nutrients through metabolic pathways. Similar to our research results, endophytic bacteria can help wheat varieties improve phosphorus utilization efficiency under low phosphorus soil conditions [67]. In addition, our results demonstrated that *Epichloë gansuensis* regulated the different metabolic pathways of amino acids to adapt to the low P concentration (0.01 mM P) and normal P concentration (0.5 mM P), respectively (Figure 8 and Figure 9). Meanwhile, *Epichloë gansuensis* increased the leaves and total PUE to adapt to the low P concentration (Figure 10). In conclusion, we confirmed the different roles of *Epichloë gansuensis* in leaves and roots of *Achnatherum inebrians* in the adaptation to different P concentrations. Under the background of global climate change, the results of this study provide a new idea for breeding low phosphorus tolerant forage by using endophytes in grasses to improve plant phosphorus utilization efficiency and maintain global phosphorus cycle balance.

## Figures and Tables

**Figure 1 jof-07-00390-f001:**
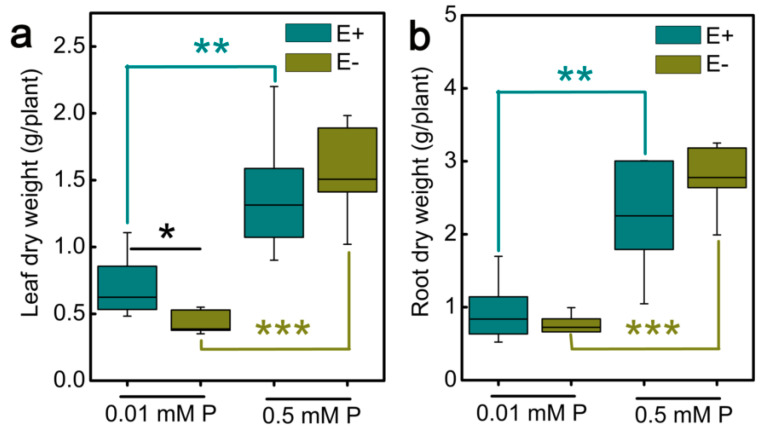
The dry weight of leaves and roots in E− and E+ plants at 0.01 and 0.5 mM P. (**a**) The dry weight of leaves; (**b**) the dry weight of roots. *, ** and *** showed differences at *p* < 0.05, *p* < 0.01, and *p* < 0.001, respectively.

**Figure 2 jof-07-00390-f002:**
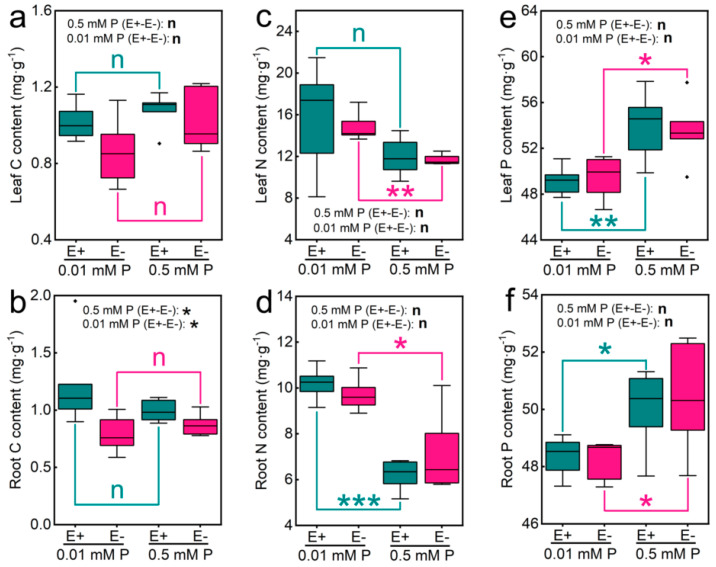
The content of C, N, and P of leaves (**a**,**c**,**e**) and roots (**b**,**d**,**f**) in E− and E+ plants at 0.01 and 0.5 mM P. *, ** and *** showed differences at *p* < 0.05, *p* < 0.01, and *p* < 0.001, respectively.

**Figure 3 jof-07-00390-f003:**
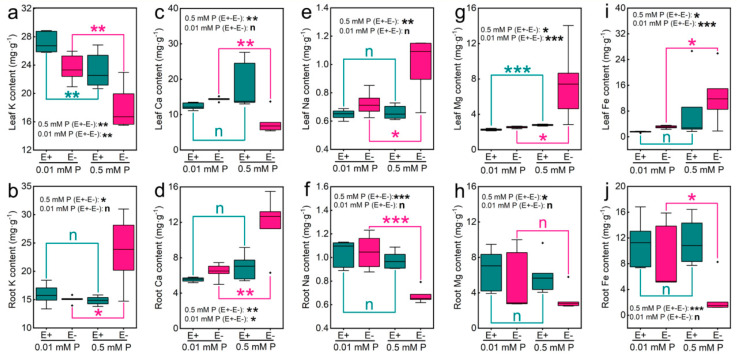
The content of K, Ca, Na, Mg, and Fe of leaves (**a**,**c**,**e**,**g**,**i**) and roots (**b**,**d**,**f**,**h**,**j**) in E− and E+ plants at 0.01 and 0.5 mM P. *, **, and *** showed differences at *p* < 0.05, *p* < 0.01, and *p* < 0.001, respectively.

**Figure 4 jof-07-00390-f004:**
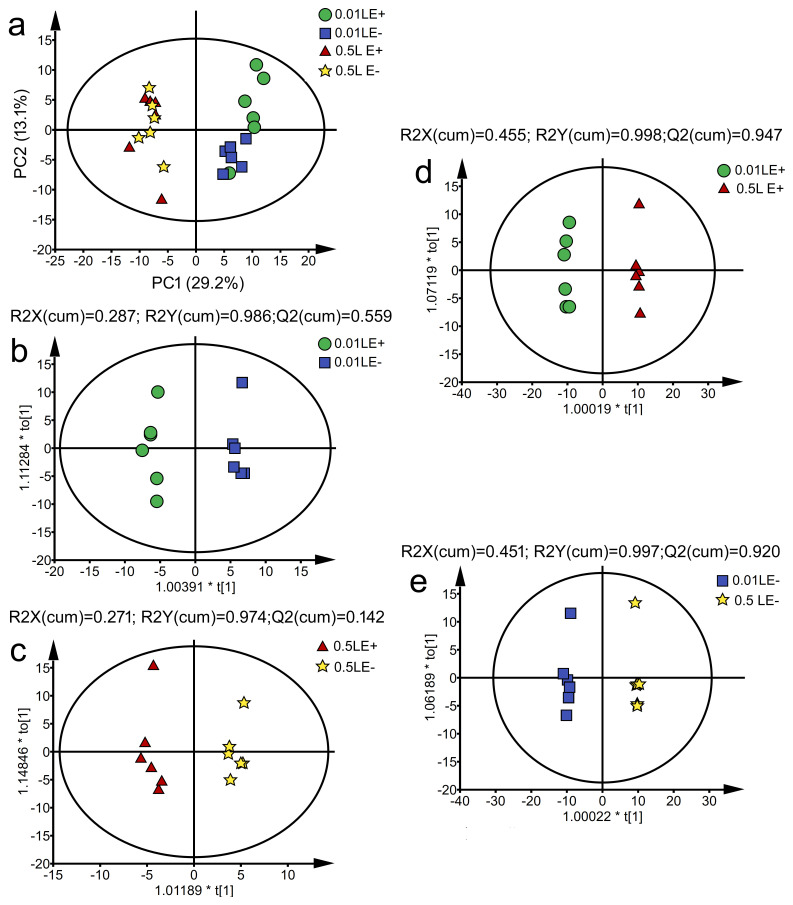
The PCA displayed the metabolic profiles in E+ and E− leaves at 0.01 and 0.5 mM P (**a**). OPLS-DA displayed *Epichloë gansuensis* effects on the metabolism of *Achnatherum inebrians* leaves between LE+ and LE− at 0.01 mM P (**b**), between LE+ and LE− at 0.5 mM P (**c**); and dose-dependence of P effects on the metabolism of *Achnatherum inebrians* leaves between 0.01 and 0.5 mM P in LE+ (**d**), between 0.01 and 0.5 mM P in LE− (**e**). LE+: E+ leaves, LE−: E− leaves.

**Figure 5 jof-07-00390-f005:**
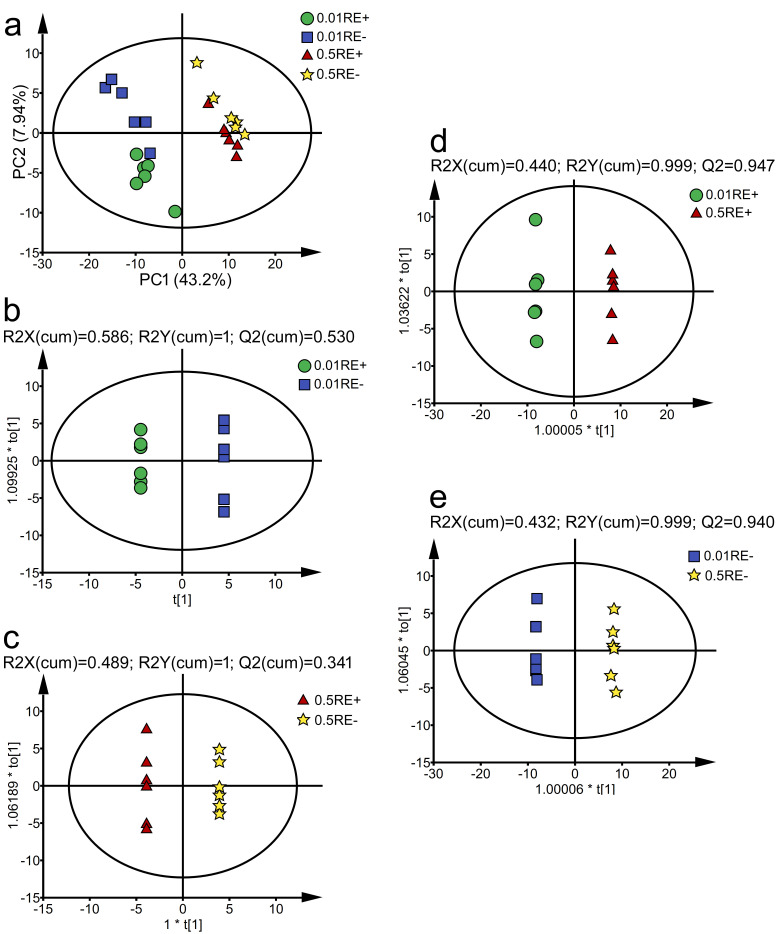
The PCA displayed the metabolic profiles in E+ and E− roots at 0.01 and 0.5 mM P (**a**). OPLS-DA displayed *Epichloë gansuensis* effects on the metabolism of *Achnatherum inebrians* roots between RE+ and RE− at 0.01 mM P (**b**), between RE+ and RE− at 0.5 mM P (**c**); and dose-dependence of P effects on the metabolism of *Achnatherum inebrians* roots between 0.01 and 0.5 mM P in RE+ (**d**), between 0.01 and 0.5 mM P in RE− (**e**). RE+: E+ roots, RE−: E− roots.

**Figure 6 jof-07-00390-f006:**
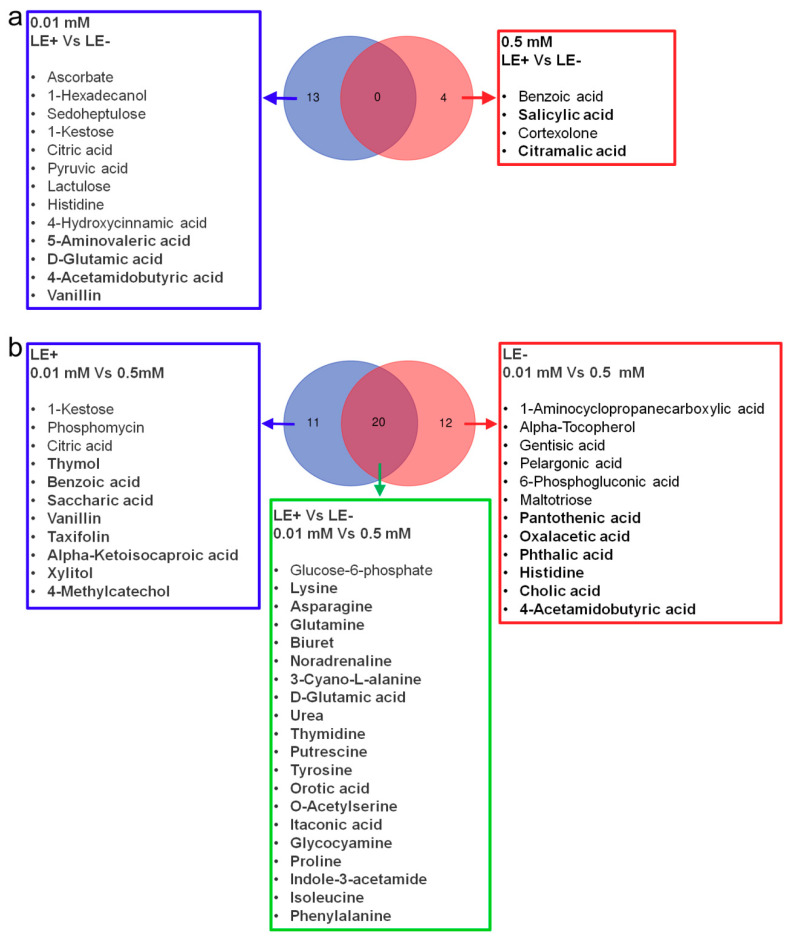
Venn diagram showed the significant difference of metabolites between LE+ and LE− at 0.01 and 0.5 mM P (**a**), between 0.01 and 0.5 mM P in LE+ and LE− (**b**).

**Figure 7 jof-07-00390-f007:**
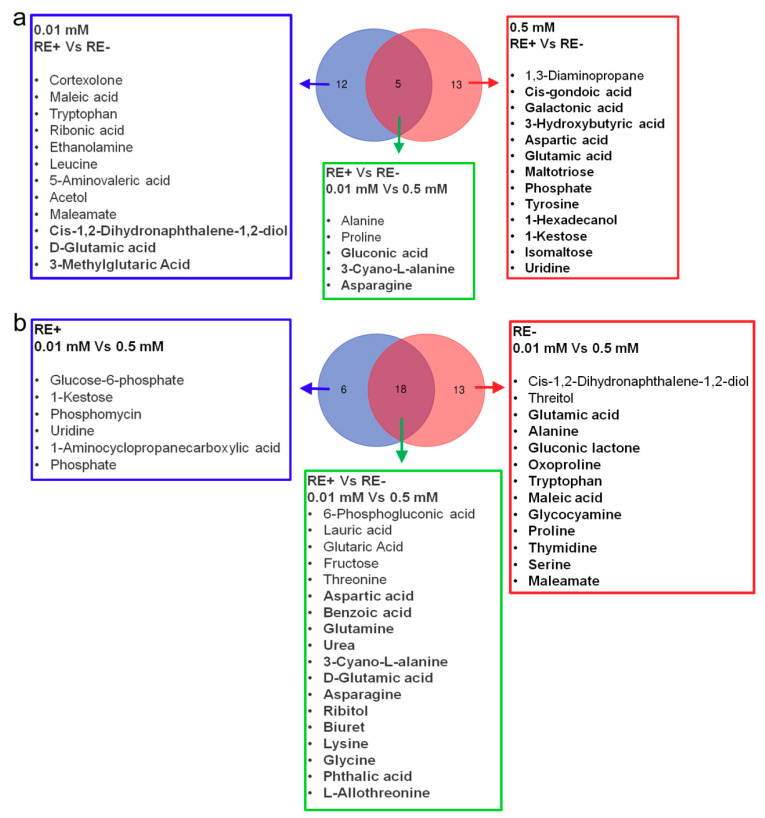
Venn diagram indicated the significant difference of metabolites between RE+ and RE− under 0.01 and 0.5 mM P (**a**), between 0.01 and 0.5 mM P in RE+ and RE− (**b**).

**Figure 8 jof-07-00390-f008:**
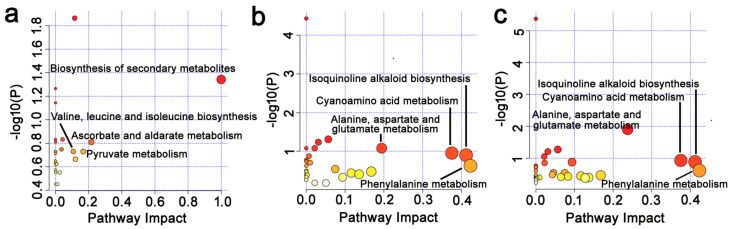
The enrichment analysis of metabolic pathway in leaves. The metabolic pathways of a, b, and c mean between LE+ and LE− at 0.01 mM P (0.01 mM LE+ vs. LE−) (**a**); between 0.01 and 0.5 mM P in LE+ (LE+ 0.01 mM vs. 0.5 mM) (**b**); between 0.01 and 0.5 mM P in LE− (LE− 0.01 mM vs. 0.5 mM) (**c**), respectively.

**Figure 9 jof-07-00390-f009:**
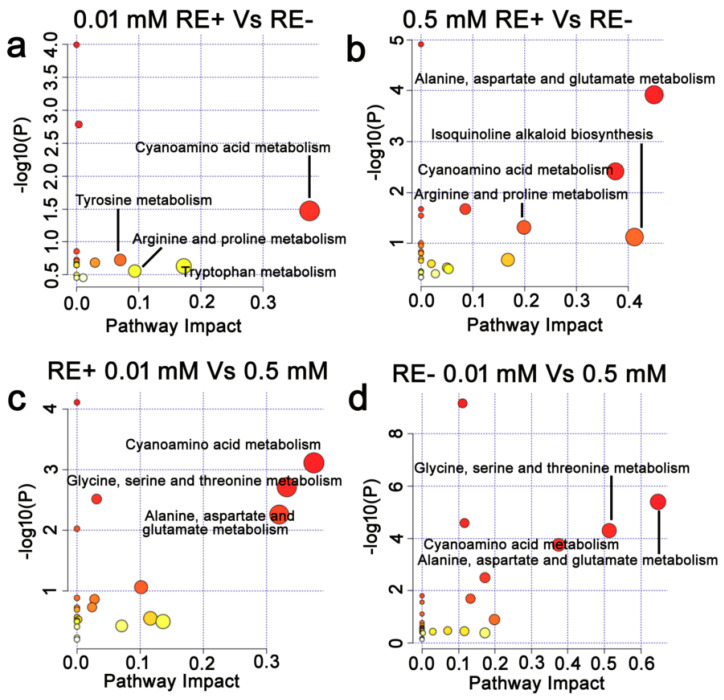
The enrichment analysis of metabolic pathway in roots. The metabolic pathways of a, b, c, and d mean between RE+ and RE− at 0.01 mM P (0.01 mM RE+ vs. RE−) (**a**); between RE+ and RE− at 0.5 mM P (0.01 mM RE+ vs. RE−) (**b**); between 0.01 and 0.5 mM P in RE+ (RE+ 0.01 mM vs. 0.5 mM) (**c**); between 0.01 and 0.5 mM P in RE− (RE− 0.01 mM vs. 0.5 mM) (**d**), respectively.

**Figure 10 jof-07-00390-f010:**
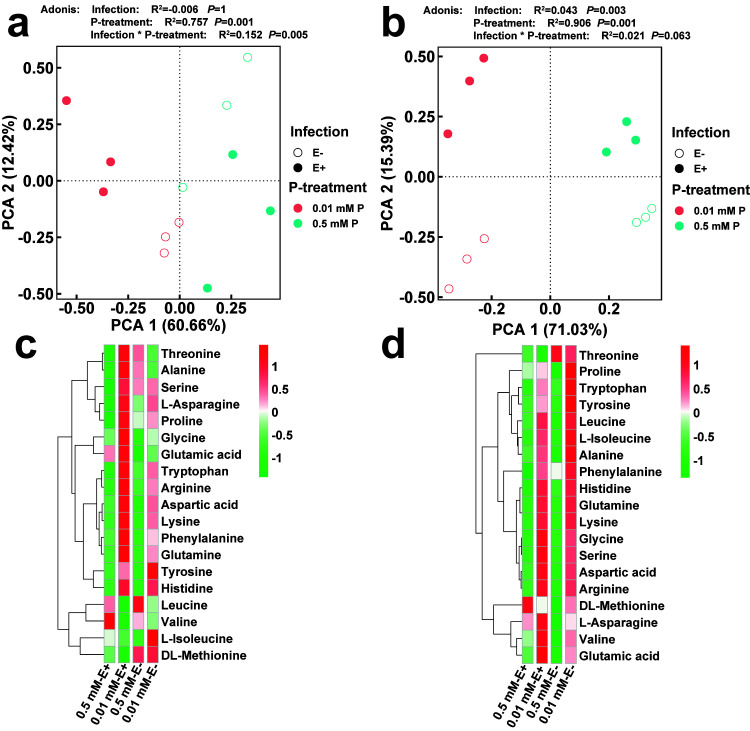
Principal component analysis for *Epichloë gansuensis*-infection and the different P concentration treatment with the content of amino acids in leaves (**a**) and roots (**c**). The content of amino acids with heatmap in leaves (**b**) and roots (**d**).

**Figure 11 jof-07-00390-f011:**
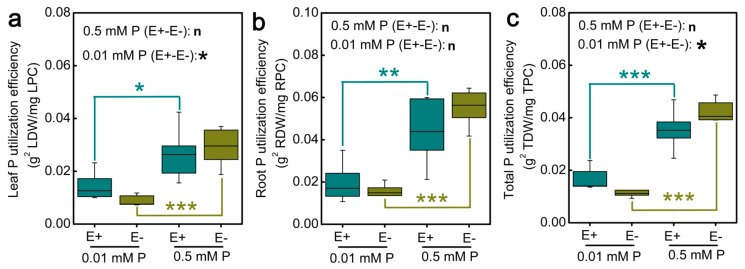
Phosphorus utilization efficiency of leaves (**a**), roots (**b**), and total (**c**) in E+ and E− *Achnatherum inebrians* at different P concentration. *, ** and *** showed differences at *p* < 0.05, *p* < 0.01, and *p* < 0.001, respectively.

## Data Availability

Not applicable.

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
