# Peer review of "Epichloë gansuensis Increases the Tolerance of Achnatherum inebrians to Low-P Stress by Modulating Amino Acids Metabolism and Phosphorus Utilization Efficiency"

_jof, 2021, doi:10.3390/jof7050390_

Round 1

Reviewer 1 Report

This manuscript "Elucidating the mechanisms for the fungal endophyte Epichloë gansuensis increases the tolerance of Achnatherum inebrians to low-P stress by regulating content and metabolic pathway of amino acids, and enhancing phosphorus utilization efficiency" is fairly well written but it needs some revisions before it is published. The comments are annotated in the manuscript. Authors should get a native English speaker to correct English. In addition, the title should be shortened. 

Author Response

Dear Editor,

We have revised the manuscript “Elucidating the mechanisms for the fungal endophyte Epichloë gansuensis increases the tolerance of Achnatherum inebrians to low-P stress by regulating content and metabolic pathway of amino acids, and enhancing phosphorus utilization efficiency” (jof-1215289) according to the reviewer’s comments. In our point-by-point responses attached below, reviewers’ comments are in black font and our responses are in blue font.

First of all, thank you very much for your great help, patient guidance and good suggestions regarding our manuscript. In our point-by-point responses below, the Editor’s and reviewer’s comments are in black font and our responses are in blue font.

Reviewer #1: This manuscript "Elucidating the mechanisms for the fungal endophyte Epichloë gansuensis increases the tolerance of Achnatherum inebrians to low-P stress by regulating content and metabolic pathway of amino acids, and enhancing phosphorus utilization efficiency" is fairly well written but it needs some revisions before it is published. The comments are annotated in the manuscript. Authors should get a native English speaker to correct English. In addition, the title should be shortened.

Firstly, thank you very much for your great help and valuable suggestion for my manuscript. We have carefully examined our manuscript, and grammar and spelling errors were found, now it had been corrected, and the native English speaker (our co-author) improved writing quality. And, we have made modifications and corrections one by one according to the modification requirements in peer-review-11844802.v1.pdf. We hope that the quality of the writing of this manuscript is now satisfactory. The title was changed into “Elucidating the mechanisms for Epichloë gansuensis increases the tolerance of Achnatherum inebrians to low-P stress by regulating metabolic pathway of amino acids and phosphorus utilization efficiency”.

Reviewer's Responses to Questions

  1. Does the introduction provide sufficient background and include all relevant references?

Reviewer #1: Can be improved

Thank you for your great help and good suggestion, I have made certain modifications and improvements in this part. I hope that the requirements can now be fully met.

  1. Is the research design appropriate?

Reviewer #1: Can be improved

Thank you.

  1. Are the methods adequately described?

Reviewer #1: Yes

  1. Are the results clearly presented?

Reviewer #1: Can be improved

Thank you, I have made certain modifications and improvements in this part. I hope that the requirements can now be fully met.

  1. Are the conclusions supported by the results?

Reviewer #1: Yes

Thank you.

  1. English language and style

Reviewer #1: Moderate English changes required

Yes, I have made correction on grammar and spelling errors, and the writing quality has been improved. Hope that the English language of the revised manuscript now can meet the requirements.

In the end, we thank the editor and the two reviewers again and hope that the revised manuscript is now acceptable.

Best regards.

Reviewer 2 Report

The manuscript entitled “Elucidating the mechanisms for the fungal endophyte Epichloë gansuensis increases the tolerance of Achnatherum inebrians to low-P stress by regulating content and metabolic pathway of amino acids, and enhancing phosphorus utilization efficiency” by Liu et al. is interesting and worth publishing.

Suggestions and corrections

  • The findings of the study may be combined and can propose a model of endophyte role on soil P- content. This will be good coverage for the readers to have a take-home message of mechanism and comprehensive ideas.
  • The author has succeeded in explaining the facts found in the present study but not in interlinking the findings with their mechanisms with respect to P and how they are increasing/decreasing?
  • In figure 1, dry weight (DW) of leaf-E+ is less than E- at 0.5 mM P concentration, what is the reason and how DW is reduced when treated with endophyte, is there any endophyte involvement in the P- content at higher concentration.
  • Line 477-480, authors described that the presence of gansuensis (endophyte) in A. inebrians plants enhanced the dry weight of leaves at the low-P stress. After few lines (483-484) described as it is decreased. It is confusing the readers, finally the endophyte is increasing or decreasing the DW?
  • A similar study (ref 26) also showed the increased root and leaf DW at low P-concentration but the authors not mentioned the exact concentration of P in the referred study.
  • The author has explained (ref 40) that low P can change the carbon metabolism. But in the present study endophyte has no significant role on carbon content/metabolism at low and/or high P, and suggest the present findings with suitable reference.
  • The author also said that carbon metabolism is linked with organic acid metabolism (568-569), according to this sentence, if C metabolism is unchanged organic acid metabolism is also should be unchanged. But in the present study it does not happen so please make it clear.

Specific comments

  • Line 20- Remove word “content” as it repeated
  • Line 22-23 and line 90-91 and throughout the manuscript- Remove word “alanine” as it repeats, what it denotes “..E+ inebrians..” What is the meaning of ..E.. mention in the first, it would be better do not to use short forms in abstract
  • Line 24-25 is not clear, please re-write
  • Line 45, 46- use the small letter at the beginning of a word, Phosphate, Unfortunately,
  • Line 59-60 – Include cultivation or distribution of this plant species in other parts of the world.
  • Lines 53-58 can be added with the last paragraph of the introduction or mention along with metabolomics, as it is jumbling the content.
  • Give gap between unit and value, for example, 27°C.
  • Line 115- “The dried leaves and roots were crushed, respectively, in a ball grinder.” not clear, reword it.
  • Line 134- remove keep in bracket “…Eppendorf.”
  • Line 388-393 are confusing, please re-write.
  • Line 515- Give a reference.
  • Line 51-58, Please change the word format and size according to the journal. guidelines and check throughout the manuscript.

The manuscript is recommended for publication after addressing the comments.

Author Response

Dear Editor,

We have revised the manuscript “Elucidating the mechanisms for the fungal endophyte Epichloë gansuensis increases the tolerance of Achnatherum inebrians to low-P stress by regulating content and metabolic pathway of amino acids, and enhancing phosphorus utilization efficiency” (jof-1215289) according to the reviewer’s comments. In our point-by-point responses attached below, reviewers’ comments are in black font and our responses are in blue font.

First of all, thank you very much for your great help, patient guidance and good suggestions regarding our manuscript. In our point-by-point responses below, the Editor’s and reviewer’s comments are in black font and our responses are in blue font.

Reviewer #2: The manuscript entitled “Elucidating the mechanisms for the fungal endophyte Epichloë gansuensis increases the tolerance of Achnatherum inebrians to low-P stress by regulating content and metabolic pathway of amino acids, and enhancing phosphorus utilization efficiency” by Liu et al. is interesting and worth publishing.

Firstly, thank you very much for your interest in the research content of the manuscript and your valuable suggestion for our manuscript. We have carefully read our manuscript and have revised the contents to make the manuscript clearer for the readers. Grammar and spelling errors have been corrected, and the writing quality has been improved.

Suggestions and corrections

The findings of the study may be combined and can propose a model of endophyte role on soil P- content. This will be good coverage for the readers to have a take-home message of mechanism and comprehensive ideas.

Thank you very much for your attention and affirmation of the research content of the manuscript and your valuable suggestions for my contributions.

The author has succeeded in explaining the facts found in the present study but not in interlinking the findings with their mechanisms with respect to P and how they are increasing/decreasing?

Firstly, thank you very much for your interest in the research content of the manuscript and your valuable suggestions for our manuscript. One study also show that endophyte promoted the biomass, phosphorus absorption and photosynthesis of inoculated maize by enhancing phosphorus enzyme activities in the rhizosphere and by decreasing the pH of the rhizosphere compared with non-inoculated controls, either in sufficient or deficient phosphorus conditions in pot cultures (Xu et al. 2020). Besides, only under stress conditions (such as low phosphorus conditions) can endophyte help plants increase phosphorus utilization efficiency and regulate their own consumption and competition of nutrients through metabolic pathways. Similar to our research results, endophytic bacteria can help wheat varieties improve phosphorus utilization efficiency under low phosphorus soil conditions (Somayeh et al. 2020). In addition, our results demonstrated that Epichloë gansuensis regulated the different metabolic pathways of amino acids to adapt to the low P concentration (0.01 mM P) and normal P concentration (0.5 mM P), respectively (Fig. 8 and Fig. 9). Meanwhile, Epichloë gansuensis increased the leaves and total PUE to adapt to the low P concentration (Fig. 10). Summary, elucidating the mechanisms for Epichloë gansuensis increases the tolerance of Achnatherum inebrians to low-P stress by regulating content and metabolic pathway of amino acids, and enhancing phosphorus utilization efficiency. The supplementary references are as follows:

Somayeh, E.; Hossein, A. A.; Ahmad, A. P.; Hassan, E.;Babak M.; Fereydoon S. Consortium of endophyte and rhizosphere phosphate solubilizing bacteria improves phosphorous use efficiency in wheat cultivars in phosphorus deficient soils. Rhizosphere. 2020, 14, 100196.

Xu, R.B.; Li, T.; Shen, M.; Yang, Z.L.; Zhao, Z.W. Evidence for a dark septate endophyte (Exophiala Pisciphila, H93) enhancing phosphorus absorptionby maize seedlings. Plant Soil. 2020 452, 249–266.

In figure 1, dry weight (DW) of leaf-E+ is less than E- at 0.5 mM P concentration, what is the reason and how DW is reduced when treated with endophyte, is there any endophyte involvement in the P- content at higher concentration.

We have already supplemented the relevant contents of this part in Discussion to explain your question: “Previous studies have shown that the effect of Neotyphodium (Epichloë) coenophialum-endophyte on the growth of tall fescue depends on soil P content, and show a more active role in high P conditions (25 mg kg-1 soil) compared with normal P supply level (5 mg kg-1 soil) [26]. In present study, dry weight of E+ plants decreased under 0.5 mM P and endophytes may lead to a certain cost of host plant growth, such as competing with host plants for nutrients and photosynthetic products. And different treatment time, host plant genotype and endophyte species all have an impact on the research results. Chen et al found that the dry weight of E+ wild barley was lower than that of E- without NaCl stress (Chen et al. 2018). Therefore, Epichloë endophyte mainly increases dry weight by improving the photosynthetic and root morphology to adapt to low phosphorus stress (Zhou 2019)”. Importantly, the dry weight of leaves and roots had no significant difference between E+ plants and E- plants, respectively. The supplementary references are as follows:

Chen, T.; Li, C.; White, J. F.; Nan, Z. Effect of the fungal endophyte Epichloë bromicola on polyamines in wild barley (hordeum brevisubulatum) under salt stress. Plant Soil. 2018, 436, 29-48.

Zhou, J. L. Interactions of nitrogen and phosphorus supply and Epichloë bromicola on growth of wild barley. Thesis of Master, Lanzhou University. 2019.

Line 477-480, authors described that the presence of gansuensis (endophyte) in A. inebrians plants enhanced the dry weight of leaves at the low-P stress. After few lines (483-484) described as it is decreased. It is confusing the readers, finally the endophyte is increasing or decreasing the DW?

Firstly, thank you very much for your valuable suggestion. I am so sorry for the confusion caused by our unclear description. We have rewritten this part. We hope that the description now is satisfactory.

A similar study (ref 26) also showed the increased root and leaf DW at low P-concentration but the authors not mentioned the exact concentration of P in the referred study.

I am so sorry for our negligence. We have supplemented the references for exact concentration of P.

The author has explained (ref 40) that low P can change the carbon metabolism. But in the present study endophyte has no significant role on carbon content/metabolism at low and/or high P, and suggest the present findings with suitable reference.

Firstly, thank you very much for your patient and valuable suggestion.Ref 40 refers to that low phosphorus stress can change carbon metabolism, but it does not involve endophyte, meanwhile, we also added the relevant content about the relationship between low phosphorus stress and carbon metabolism: “Our study showed that although low phosphorus stress did not significantly affect the carbon content of leaves and roots, it changed the carbon metabolism, including the metabolite content of some sugars and organic acids (SI. Table 1 and 2) and amino acid metabolic pathway (Fig. 8 and 9)”.

The author also said that carbon metabolism is linked with organic acid metabolism (568-569), according to this sentence, if C metabolism is unchanged organic acid metabolism is also should be unchanged. But in the present study it does not happen so please make it clear.

Firstly, thank you very much for your patient and valuable suggestion.What we want to clarify in this part of the discussion section is: C metabolism mainly includes organic acid metabolism, sugar metabolism and metabolism related to photosynthetic enzymes. As a part of the process of C metabolism, the change of organic acid metabolism will lead to the change of C metabolism, the change of C metabolism is not necessarily caused by organic acid metabolism. In the present study, low P stress and endophyte had no significant effect on the C content of the whole plant and leaves, respectively, they changed the C metabolism by changing the metabolite content of some sugars and organic acids (SI. Table 1 and 2) and amino acid metabolic pathway (Fig. 8 and 9).

List of some specific comments:

Line 20- Remove word “content” as it repeated

Yes, I have removed the repeated word “content”.

Line 22-23 and line 90-91 and throughout the manuscript- Remove word “alanine” as it repeats, what it denotes “..E+ inebrians..” What is the meaning of ..E.. mention in the first, it would be better do not to use short forms in abstract

I have removed the repeated word “alanine”. And I have added the meaning of “E+ / E− inebrians” in the first and replaced them with their full name in abstract.

Line 24-25 is not clear, please re-write

Yes, I have rewritten this sentence.

Line 45, 46- use the small letter at the beginning of a word, Phosphate, Unfortunately,

Yes, I have corrected the wrong word case.

Line 59-60 - Include cultivation or distribution of this plant species in other parts of the world.

Thank you very much for your good suggestions.

Lines 53-58 can be added with the last paragraph of the introduction or mention along with metabolomics, as it is jumbling the content.

Yes, I have rewritten this sentence.

Give gap between unit and value, for example, 27°C.

Yes, I have revised it.

Line 115- “The dried leaves and roots were crushed, respectively, in a ball grinder.” not clear, reword it.

Yes, I have rewritten this sentence.

Line 134- remove keep in bracket “…Eppendorf.”

Yes, I have revised it.

Line 388-393 are confusing, please re-write.

Yes, I have rewritten this sentence.

Line 515- Give a reference.

Yes, I have provided the corresponding reference.

Line 51-58- Please change the word format and size according to the journal guidelines and check throughout the manuscript.

I am so sorry for my carelessness. I have revised the word format and size in the entire manuscript according to the journal guidelines.

Reviewer's Responses to Questions

  1. Does the introduction provide sufficient background and include all relevant references?

Reviewer #2: Can be improved

Thank you for your great help and good suggestion, I have made certain modifications and improvements in this part. I hope that the requirements can now be fully met.

  1. Is the research design appropriate?

Reviewer #2: Yes

Thank you.

  1. Are the methods adequately described?

Reviewer #2: Yes

Thank you.

  1. Are the results clearly presented?

Reviewer #2: Can be improved

Thank you, I have made certain modifications and improvements in this part. I hope that the requirements can now be fully met.

  1. Are the conclusions supported by the results?

Reviewer #2: Can be improved

Thank you, I have made certain modifications and improvements in this part. I hope that the requirements can now be fully met.

  1. English language and style

Reviewer #2: Moderate English changes required

Yes, I have made correction on grammar and spelling errors, and the writing quality has been improved. Hope that the English language of the revised manuscript now can meet the requirements.

In the end, we thank the editor and the two reviewers again and hope that the revised manuscript is now acceptable.

Best regards.